

# When safety becomes the priority: defensive nursing practice and its associated factors among nurses in Egypt: a cross-sectional study

Ahmed Zaher[1,*], Yasmine M. Osman[2], Salwa Sayed[3], Sally Mohammed Farghaly Abdelaliem[4], Amany Anwar Saeed Alabdullah[5], Ahmed Hendy[6,7], Zainab Attia Abdallah[8], Mohammed Musaed Ahmed Al-Jabri[9] and Abdelaziz Hendy[10,*]

[1] Psychiatric Mental Health Nursing, Faculty of Nursing, Ain Shams University, Cairo, Egypt
[2] Department of Obstetrics and Gynecology Nursing, Faculty of Nursing, Zagazig University, Zagazig, Egypt
[3] Technical Health Institute, General Authority for Health Insurance, Cairo, Egypt
[4] Department of Nursing Management and Education, College of Nursing, Princess Nourah bint Abdulrahman University, Riyadh, Saudi Arabia
[5] Department of Maternity and Pediatric Nursing, College of Nursing, Princess Nourah bint Abdulrahman University, Riyadh, Saudi Arabia
[6] Department of Computational Mathematics and Computer Science, Institute of Natural Sciences and Mathematics, Ural Federal University, Yekaterinburg, Russia
[7] Department of Mechanics and Mathematics, Western Caspian University, Baku, Azerbaijan
[8] Community Health Nursing, Faculty of Nursing—Modern University for Technology and Information (MTI), Cairo, Egypt
[9] Critical Care Nursing, Prince Sattam bin Abdulaziz University, College of Applied Medical Sciences, Nursing Department, Wadi Aldawaser, Saudi Arabia
[10] Pediatric Nursing Department, Faculty Nursing Ain Shams University, Cairo, Egypt
* These authors contributed equally to this work.

Corresponding author
Abdelaziz Hendy,
Abdelaziz.hendy@nursing.asu.edu.eg

## ABSTRACT

**Background.** Defensive nursing practices, which prioritize legal protection over patient care, are becoming increasingly common. This study aims to explore the prevalence and factors associated with defensive nursing practices among nurses in Egypt, considering the impact of workplace violence and legal threats.

**Methods.** A descriptive cross-sectional study was conducted from February to April 2024 using a self-report online questionnaire. The target population included clinical nurses working in various hospitals in Egypt. A sample size of 1,267 nurses was achieved through convenience sampling. The questionnaire assessed demographic data, experiences of workplace violence, legal consequences, and defensive nursing practices, categorized into positive and negative behaviors.

**Results.** The sample consisted of 1,267 nurses, predominantly female (75.9%), with a mean age of 28.57 years. Positive defensive practices, such as detailed documentation (79%) and thorough explanation of procedures (58.5%), were highly prevalent. Negative practices included avoiding high-risk procedures (15.9%) and patients more likely to file lawsuits (13.6%). Older nurses and those with higher educational qualifications were more likely to engage in positive defensive practices. Nurses who experienced workplace violence or legal threats were significantly more likely to avoid high-complication procedures.

**Conclusion**. The study identified a high engagement in both positive and negative defensive practices among nurses in Egypt. These practices are influenced by factors such as age, education level, and experiences of workplace violence and legal threats. The findings underscore the need for strategies to support nurses, reduce reliance on defensive practices, and ensure better patient outcomes.

# INTRODUCTION

Globally, millions of people require complex health and social care services, creating significant challenges and opportunities for healthcare systems (*Nyashanu, Pfende & Ekpenyong, 2020*). These challenges include limited financial resources, increased population density, rising rates of comorbid chronic diseases, and rapid technological advancements. Addressing these issues requires the provision of comprehensive and seamless care, which remains a central concern for nurses. As integral members of healthcare teams, nurses play a critical role in ensuring patient safety by mitigating risks and preventing adverse events (*Han, Kim & Seo, 2020*).

Technological advancements, enhanced understanding of diseases, and increasing demands in clinical environments have transformed the traditional patient-nurse relationship (*Boyne et al., 2022*). As a result, patients and their families now hold higher expectations of nurses, including the provision of comprehensive educational and support services as well as active involvement in care decisions (*Hannawa et al., 2022*). While these developments aim to enhance patient care, they may inadvertently lead to challenges, such as restrictive practices that threaten patient safety (*Berzins et al., 2020*).

In parallel with the expansion of healthcare services, the legal dimensions of nursing practice have become more pronounced, reflecting similar trends globally (*Parker, 2002*; *Pazvantoğlu et al., 2011*). In this context, nurses may resort to defensive practices to avoid errors and disciplinary actions, often feeling pressured to meet organizational expectations and goals (*Manuel & Crowe, 2014*). Over the past few decades, the culture of defensive practice has grown significantly, driven by an increasing number of malpractice claims and legal threats, particularly in high-risk medical areas (*Tuers, 2013*). Defensive practices are not exclusive to physicians but are also prevalent among other healthcare professionals, such as nurses and midwives (*Mullen, Admiraal & Trevena, 2008*; *Rinaldi et al., 2019*). This phenomenon occurs when practitioners prioritize self-protection over patient interests due to the fear of liability claims and lawsuits (*Baungaard et al., 2020*). While defensive practices aim to mitigate risks, they can lead to unintended consequences, such as suboptimal patient care and hindrances to evidence-based practice implementation (*Rothberg et al., 2014*). Furthermore, defensive practices can exacerbate malpractice cases and infringe upon patients' rights, raising ethical and practical concerns (*Renkema, Broekhuis & Ahaus, 2014*).

Defensive nursing practices can be broadly categorized into positive and negative behaviors. Positive defensive practices include actions such as detailed documentation and thorough explanations to patients, which enhance transparency, accountability, and patient trust (*Moosazadeh et al., 2014*; *Ortashi et al., 2013*). These practices are often aimed at reducing the risk of malpractice litigation while maintaining high standards of care (*Miziara & Miziara, 2022*). In contrast, negative defensive practices are strategies employed to minimize personal legal risks but may compromise patient care. Examples include avoiding high-risk procedures or patients perceived as litigious, refusing to perform invasive treatments, and excluding high-risk patients from surgical schedules (*Arafa et al., 2023*). These behaviors, while protective for the practitioner, can adversely impact patients' access to necessary medical interventions and may lead to disparities in care delivery (*O'Connell, 2021*). For instance, a prior study reported that 12% of midwives ceased offering or attending vaginal births after cesarean sections due to fear of liability and litigation. This highlights how defensive practices, particularly negative ones, can create barriers to evidence-based care and patient safety (*Guidera et al., 2012*).

Over the past five years, the Egyptian healthcare system has undergone significant reforms and the implementation of new regulations, including the adoption of the landmark Egyptian Universal Health Insurance (UHI) law (*WHO, 2024*). These changes have been widely perceived as positive steps toward improving healthcare quality and accessibility. However, despite these advancements, the prevalence of defensive practices has increased, and data regarding the scope and impact of these practices remain insufficient.

Nurses in Egypt face considerable challenges, particularly regarding legal liabilities associated with nursing malpractice cases, with perceptions of these liabilities varying widely among practitioners (*Mahmoud Farhat, Ghandour & Mohamed, 2023*). Additionally, workplace violence (WPV) represents a major occupational hazard for healthcare professionals, including nurses. Patients and their relatives are often the primary perpetrators, with incidents ranging from verbal abuse to physical assaults (*Abozaid et al., 2022*). WPV has a detrimental effect on nurses, impeding their ability to provide essential care and significantly compromising the overall quality of healthcare delivery (*Cheung & Yip, 2017*; *Kafle et al., 2022*).

To address these challenges, it is essential to increase nurses' awareness of defensive practices, address resource limitations, and promote workplace safety measures. Moreover, enhancing nurses' autonomy in their roles can contribute to reducing the reliance on defensive behaviors and improving their ability to deliver patient-centered care (*Bowers et al., 2015*).

To date, no studies have explored the prevalence and factors contributing to defensive nursing practices in Egypt. This study addresses this critical gap by examining the individual, social, organizational, and ethical dimensions of defensive nursing. By identifying these factors, evidence-based strategies can be proposed to reduce reliance on defensive practices, promote staff safety, and enhance patient care. This cross-sectional study aims to analyze the prevalence and types of defensive nursing practices among Egyptian nurses and identify the factors driving their adoption.

## METHODS

### Study design

A descriptive cross-sectional study using a self-report online questionnaire was conducted and reported in accordance with the guidelines for Strengthening the Reporting of Observational Studies in Epidemiology (STROBE) (*Von Elm et al., 2014*). The target population of this study consisted of all clinical nurses, including females and males, who were 20 years or older and currently working in affiliated universities and public or private hospitals, and willing to participate in the study. Internship nursing students who were undergoing training at the hospital were excluded from the study.

### Sampling and sample size calculation

Convenient sampling was used to conduct this study. The sample size was calculated based on a study carried out by *Turan & Kaya (2019)*, which estimated an effect size of 65.5% of nurses who reported that they always kept their records in a more detailed way to protect themselves against allegations of malpractice. The level of confidence (1-Alpha Error) was 95%, the margin of error was 5, the population proportion was 65.5, and the population size was 300,000 registered nurses in the nursing syndicate. Therefore, the minimum final sample size was determined to be 162. However, we more than quadrupled the least required sample size to allow for assessing different frequencies of defensive nursing practices.

### Tools of data collection

The data were collected using a self-administered online questionnaire. The questionnaire was developed based on an extensive review of relevant studies and previously published questionnaire instruments. It is intended to assess the practices of defensive nursing among Egyptian nurses. The questionnaire was translated into Arabic for the purpose of this research; two bilingual translators independently translated the English version into Arabic to evaluate the applicability of the defensive nursing practice tool for Arabic-speaking populations. To ensure the accuracy of the translation, two additional impartial translators back-translated the translation into English. It consisted of three parts:

**Part I:** The first part of the questionnaire gathered the demographic data of nurses, including age, gender, marital status, qualifications, years of working experience, and specialty (department).

**Part II:** Factors associated with defensive nursing practice assessed through experience of nurses-related violence and legal consequences such as experiencing physical violence in the workplace, threats by patients or their companions that you might face legal consequences related to nursing practice and experiencing legal consequences due to circumstances related to nursing practice.

**Part III:** Defensive nursing practices were assessed with ten items divided into two domains: it was adopted from *Turan & Kaya (2019)*. It included positive defensive practices such as "Do you carry out interventions or procedures that are probably not unnecessary to avoid possible legal consequences?" and "Do you order tests without a doctor's prescription that are probably not clinically indicated to avoid possible legal consequences?" *etc.*; and

negative defensive practices such as "Do you refuse to assign high-risk patients to avoid possible legal consequences in the case of complications?" and "Do you avoid high-risk procedures to avoid possible legal consequences in the case of complications?". The study utilized a Likert scale scoring system in which each item was rated as always "2", sometimes "1", and never "0", and a high score indicates a high level of defensive nursing practices. The nurses' responses were categorized into negative defensive practices if the score was from 0–5, and positive defensive practices if the score was from 6–10.

The translation process of the questionnaire followed a rigorous, standardized methodology to ensure linguistic and cultural equivalence. This included forward translation by bilingual translators, reconciliation of translations, back-translation by independent translators, expert review, and pilot testing with a small group of nurses. These steps ensured semantic, idiomatic, and conceptual alignment with the original version. The approach aligns with established best practices, as emphasized in studies such as *Thompson et al. (2024)*, which highlight the importance of nuanced and validated translation methods. By adhering to these practices, the study ensured the validity and reliability of the translated questionnaire for the target population.

### Tools validity

The researchers created the questionnaire after reviewing pertinent literature. A five-person jury of professionals from the education and psychiatric nursing departments assessed it. The jury found that the scales sufficiently evaluated the intended structures.

### Pilot study

The pilot study, which involved 10% of the nurses from the context above, evaluated the practicality and clarity of the items, identified any potential difficulties or issues that might arise during data collection, and measured the time needed to complete the tools.

### Procedures

The online questionnaire was disseminated through various social media platforms and online forums frequented by nurses in Egypt. It consisted of three sections: the first provided an overview of the study's objectives and eligibility criteria; the second gathered demographic and workplace information; and the third assessed different defensive nursing practices. The questionnaire was created using Google Forms (https://forms.office.com/r/0SA4hJEZX1). To encourage participation, the survey link was shared in several social networking groups catering to Egyptian nurses, including platforms like Facebook and WhatsApp. Additionally, the Egyptian Nursing Syndicate, the primary authority responsible for issuing nursing licenses in Egypt, helped circulate the survey within its online networks. Other national professional nursing associations and societies shared the questionnaire with their members. Even nursing students were engaged in distributing the survey forms over WhatsApp Platforms because of their continuing attendance at the hospital for training and education, and they did so after distributing the questionnaire link and invitation to nurses.

Data was collected for one month, from February 2024 to April 2024. An online informed consent form was created, and participants had to agree to it by clicking an "I agree" button

before moving on to the survey questions. Each question in the questionnaire was made mandatory, meaning that partial responses could not be submitted. The researchers closed the question page once the responses reached the estimated sample size. Finally, all completed questionnaires were printed out and entered into the statistical package (SPSS) to avoid any mistakes that might occur.

## Ethical consideration

The study received ethical clearance from the research committee at the Faculty of Nursing, Modern University for Technology and Information (MTI), under approval number FAN/128/2023, dated December 25, 2023. An online consent form was developed, requiring participants to confirm their agreement by selecting the ''I agree'' option before proceeding to the survey questions. Upon agreeing to the consent form's initial question, participants were automatically redirected to the survey. Participants were informed that their responses would remain confidential, accessible only to authorized research team members. The survey was designed to be anonymous, ensuring the privacy of participants. Participation was entirely voluntary, and nurses could opt out of the study at any stage.

## Statistical analysis

The data collected were coded and entered into the Statistical Package for the Social Sciences (SPSS) software, version 24 (IBM Corp., Armonk, NY, USA). After data entry, the dataset was thoroughly examined for potential errors and then analyzed using the same program to generate frequency tables with percentages. Qualitative data were represented as numbers and percentages, while quantitative data were described using means and standard deviations, as appropriate. The chi-square test was employed to assess the association between two categorical variables, with results considered statistically significant at $P \leq 0.05$ and highly significant at $P < 0.01$. The reliability of the developed tools was assessed using Cronbach's alpha coefficient, calculated in SPSS version 24 by a statistician. The questionnaire demonstrated good internal consistency reliability, with a Cronbach's alpha value of 0.836.

## RESULTS

The sample consisted of 1,267 nurses (75.9% female) with a mean age of 28.57 years (SD = 5.8). Regarding marital status, half were single (50%) while 44.9% were married. Most had either a bachelor's degree in nursing (45.8%) or a nursing diploma/institute degree (17.8%), with 7.4% having a postgraduate degree. The mean years of working experience was 7.6 (SD = 6.3), with 69.5% having 1–10 years of experience. The nurses specialized in various departments, including ICU (33.3%), wards (33.1%), emergency (13.3%), operation theaters (14.2%), and psychiatry (8%). Also, nearly one-fifth (19.7%) reported experiencing physical violence at work. A sizeable proportion (32%) had been threatened by patients or their companions that they might face legal consequences related to their nursing practice. Furthermore, 15.9% of the nurses had actually experienced legal consequences due to circumstances related to their nursing duties. Also, the majority (57.9%) did not view their risk as particularly high at all. However, a sizable 31.8%

considered their legal risk to be high, while 10.3% even rated their risk as profoundly high, see more in Table 1.

Table 2 outlines the practices of defensive nursing reported by the 1,267 nurses surveyed. Both positive and negative defensive practices were assessed. Positive practices like explaining nursing procedures in more detail (58.5% always did this) and keeping more detailed records (79% always) were highly prevalent, likely aimed at protecting against malpractice allegations. However, some negative defensive practices were also reported, though to a lesser extent—such as avoiding high-risk procedures (15.9% always did this) and avoiding patients more likely to file lawsuits (13.6% always).

Table 3 presents the relation between nurses' socio-demographic characteristics and the practice of explaining nursing procedures in more detail to protect against malpractice allegations. There were significant differences based on age ($\chi^2 = 18.124$, $p = .001$), educational qualifications ($\chi^2 = 32.539$, $p < .001$), and clinical specialty ($\chi^2 = 31.065$, $p < .001$). Also, nurses aged 40–50 years old (74.7%) were more likely to always explain in detail compared to those 20-< 30 years (6.6%) and 30-< 40 years (9%). Similarly, higher proportions of nurses with bachelor's (60.5%) and postgraduate degrees (51%) always did this compared to those with diplomas/institutes (48.6%). Across departments, emergency (69.1%) and psychiatry nurses (50.9%) most frequently always explained in detail. However, years of experience was not significantly associated with this practice ($\chi^2 = 3.826$, $p = .430$).

Table 4 shows that nurses who had experienced physical workplace violence were more likely to always (35.7%) or sometimes (49.8%) avoid high-complication procedures compared to those without such experiences (always 25.2%, sometimes 40.4%), $\chi^2 = 38.349$, $p < .001$. Similarly, those threatened with legal consequences by patients/families more frequently always (32.3%) or sometimes (45.2%) avoided these procedures *versus* those not threatened (always 25%, sometimes 40.8%), $\chi^2 = 19.260$, $p < .001$. Nurses who had faced legal consequences related to nursing practice were also more prone to always (33.1%) or sometimes (52%) avoid high-complication procedures compared to those without legal issues (always 26.2%, sometimes 40.4%), $\chi^2 = 27.668$, $p < .001$. Across departments, emergency nurses most commonly always (32%) avoided these procedures, $\chi^2 = 15.843$, $p = .045$. These results suggest prior violence/legal experiences significantly increase defensive tendency to avoid high-risk nursing procedures, potentially compromising care quality. According to the the relationship between nurses' perceived risk of malpractice lawsuits at work, clinical specialty, and the practice of keeping records in a more detailed way to protect against allegations. Chi-square analyses found a significant association with perceived legal risk ($\chi^2 = 9.679$, $p = .046$) and specialty department ($\chi^2 = 50.599$, $p < .001$). Nurses who rated their malpractice risk as profoundly high were most likely to always keep ultra-detailed records (81.7%) compared to those rating risk as high (80.4%) or not at all high (77.8%). Across specialties, all emergency nurses (100%) and nearly all operation room nurses (88.1%) reported always maintaining extremely thorough documentation. In contrast, lower proportions of psychiatry (74.2%) and ward nurses (77.3%) always did so.

**Table 1 Characteristics of studied nurses (n = 1,267).**

| Items | N | % |
|---|---|---|
| Age: | | |
| 20–<30 | 784 | 61.8 |
| 30–<40 | 408 | 32.2 |
| 40–50 | 75 | 6 |
| Mean ± SD 28.57 (5.8) | | |
| Gender: | | |
| Female | 962 | 75.9 |
| Male | 305 | 24.1 |
| Marital Status: | | |
| Single | 633 | 50 |
| Married | 569 | 44.9 |
| Divorced | 52 | 4.1 |
| Widowed | 13 | 1 |
| Qualifications: | | |
| Nursing Diploma | 94 | 7.4 |
| Nursing institute | 499 | 39.4 |
| Bachelor's degree | 580 | 45.8 |
| Postgraduate | 94 | 7.4 |
| Years of working experience: | | |
| 1–<10 | 881 | 69.5 |
| 10–<20 | 299 | 23.6 |
| 20–30 | 87 | 6.9 |
| Mean ± SD 7.6 (6.3) | | |
| Specialty (Department): | | |
| ICU | 422 | 33.3 |
| Ward | 396 | 31.3 |
| Emergency | 181 | 14.2 |
| Operation | 101 | 8 |
| Psychiatry | 167 | 13.2 |
| Experienced physical violence in the workplace: | | |
| Yes | 249 | 19.7 |
| No | 1,018 | 80.3 |
| Threatened by patients or their companions that you might face legal consequences | | |
| Yes | 405 | 32 |
| No | 862 | 68 |
| Experienced legal consequences due to circumstances related to nursing practice: | | |
| Yes | 202 | 15.9 |
| No | 1,065 | 84.1 |
| Risk of malpractice lawsuit at work: | | |
| Profoundly high | 131 | 10.3 |
| High | 403 | 31.8 |
| Not at all | 733 | 57.9 |

**Table 2  Practices of defensive nursing among studied nurses (*n* = 1,267).**

| The practices of defensive nursing | Never | | Sometimes | | Always | |
|---|---|---|---|---|---|---|
| | N | % | N | % | N | % |
| **Positive defensive practices** | | | | | | |
| Carry out interventions or procedures that are probably not unnecessary to avoid possible legal consequences | 659 | 52 | 513 | 40.5 | 95 | 7.5 |
| Order tests that are probably not clinically indicated without a doctor's prescription to avoid possible legal consequences | 882 | 69.6 | 620 | 25.3 | 65 | 5.1 |
| Having severe concerns about making mistakes in nursing care | 181 | 14.3 | 700 | 55.2 | 386 | 30.5 |
| Explain nursing practices in more detail to protect yourself from malpractice allegations | 92 | 7.3 | 434 | 34.3 | 741 | 58.5 |
| Keep the records in a more detailed way to protect yourself from malpractice allegations | 55 | 4.3 | 211 | 16.7 | 1,001 | 79 |
| **Negative defensive practices** | | | | | | |
| Refuse to assign high-risk patients to avoid possible legal consequences in the case of complications | 804 | 63.5 | 362 | 28.5 | 101 | 8 |
| Avoid high-risk procedures to avoid possible legal consequences in the case of complications | 522 | 41.2 | 544 | 42.9 | 201 | 15.9 |
| Administer drugs that you think to be unnecessary to protect yourself from malpractice allegations | 875 | 69.1 | 255 | 20.1 | 137 | 10.8 |
| Avoid patients who are more likely to file a lawsuit to protect yourself from malpractice allegations | 632 | 49.9 | 462 | 36.5 | 173 | 13.6 |
| Avoid practices with high complications to protect yourself from malpractice allegations | 386 | 30.5 | 535 | 42.2 | 346 | 27.3 |

## DISCUSSION

This study delves into the explore and analyze the prevalence and types of defensive nursing practices among nurses in Egypt. Also, identify the factors that contribute to the adoption of these practices. By exploring that, valuable insights can be gained to improve nursing practices and optimize patient safety in healthcare settings in Egypt. The current study examined the prevalence of workplace violence and the legal consequences encountered by 1,267 nurses. The results revealed that a significant proportion of the nurses reported experiencing physical violence in their workplace. Additionally, a considerable number of respondents reported being threatened by patients or their companions with potential legal consequences as a result of circumstances linked to their nursing duties.

To ensure the robustness of our findings, we increased the sample size to more than seven times the minimum required. This approach allowed us to assess various frequencies and patterns of defensive nursing practices comprehensively. Several strategies were employed to achieve high participation rates among nurses. The survey link was widely disseminated through professional nursing associations, social media platforms, and workplace networks. Additionally, leveraging the support of organizations such as the Egyptian Nursing Syndicate enhanced outreach to diverse nursing professionals across various specialties and regions. The relevance of defensive nursing practices as a pressing

**Table 3 Relation between nurses' characteristics and explaining nursing practice in more details.**

| Socio- demographic data | | Explain nursing practices in more detail to protect yourself from malpractice allegations | | | | | | $X^2$ | P- value |
|---|---|---|---|---|---|---|---|---|---|
| | | Always (n = 92) | | Sometimes (n = 434) | | Never (n = 741) | | | |
| | | No. | % | No. | % | No. | % | | |
| Age (years) | 20–<30 | 475 | 60.6 | 257 | 32.8 | 52 | 6.6 | 18.124 | .001 |
| | 30–<40 | 210 | 51.5 | 161 | 39.5 | 37 | 9 | | |
| | 40–50 | 56 | 74.7 | 16 | 21.3 | 3 | 4 | | |
| Qualifications | Nursing Diploma | 40 | 42.6 | 48 | 51 | 6 | 6.4 | 32.539 | <.001 |
| | Nursing institute | 310 | 62.1 | 161 | 32.3 | 28 | 5.6 | | |
| | Bachelor's degree | 351 | 60.5 | 177 | 30.5 | 52 | 9 | | |
| | Postgraduate | 40 | 42.6 | 48 | 51 | 6 | 6.4 | | |
| Years of work-ing experience | 1–<10 | 510 | 58.1 | 307 | 35 | 61 | 6.9 | 3.826 | .430 |
| | 10–<20 | 172 | 57.5 | 102 | 34.1 | 25 | 8.4 | | |
| | 20–30 | 56 | 66.7 | 25 | 29.8 | 3 | 3.5 | | |
| Specialty (Department) | Emergency | 125 | 69.1 | 43 | 23.8 | 13 | 7.2 | 31.065 | <.001 |
| | ICU | 256 | 60.7 | 152 | 36 | 14 | 3.3 | | |
| | Operation | 57 | 56.4 | 38 | 37.6 | 6 | 5.9 | | |
| | Psychiatry | 85 | 50.9 | 63 | 37.7 | 19 | 11.4 | | |
| | Ward | 218 | 55.1 | 138 | 34.8 | 40 | 10.1 | | |

issue in Egypt further contributed to participant engagement. Nurses face significant challenges, including workplace violence and legal risks, making the study topic highly pertinent to their professional experiences. This relevance likely motivated many nurses to share their insights, enriching the study with a wide range of perspectives.

The findings of this study are consistent with existing literature, highlighting the pervasive nature of workplace violence in healthcare settings. Previous studies have shown that nurses are particularly vulnerable to various forms of violence, including physical, verbal, and psychological abuse (*Öztaş, Yava & Koyuncu, 2023*). This study adds to the body of evidence by quantifying the extent of physical violence and legal threats faced by nurses. The high incidence of physical violence reported by nurses is alarming and indicates a critical need for improved safety measures in healthcare facilities (*Yenealem & Mengistu, 2024*). Physical violence can lead to severe physical injuries, psychological trauma, and long-term health issues for the victims (*Powell et al., 2023*). The threat of legal consequences as reported by the nurses is a significant concern (*Salehi et al., 2021*). These threats can stem from misunderstandings, dissatisfaction with care, or attempts to intimidate healthcare professionals. The fear of legal repercussions can lead to defensive medicine practices, increased stress, and burnout among nurses, further exacerbating the challenges they face (*Ries, Johnston & Jansen, 2022*).

Our study indicates that more than one third of nurses consider their risk of facing a malpractice lawsuit to be high. This perception can significantly impact their work behavior

**Table 4 Relation between nurses' experience of violence, legal consequences, avoidance of high-complication practices, and detailed record keeping.**

| Experience of nurses-related violence and legal consequences | | Avoid practices with high complications to protect yourself from malpractice allegations | | | | | | $X^2$ | $P$-Value |
|---|---|---|---|---|---|---|---|---|---|
| | | Always (n = 346) | | Sometimes (n = 535) | | Never (n = 386) | | | |
| | | No. | % | No. | % | No. | % | | |
| Experienced physical violence in the workplace | No | 257 | 25.2 | 411 | 40.4 | 350 | 34.4 | 38.349 | <.001 |
| | Yes | 89 | 35.7 | 124 | 49.8 | 36 | 14.5 | | |
| Threatened by patients or their companions that you might face legal consequences | No | 215 | 25 | 352 | 40.8 | 295 | 34.2 | 19.260 | <.001 |
| | Yes | 131 | 32.3 | 183 | 45.2 | 91 | 22.5 | | |
| Experienced legal consequences due to circumstances related to nursing practice | No | 279 | 26.2 | 430 | 40.4 | 356 | 33.4 | 27.668 | <.001 |
| | Yes | 67 | 33.1 | 105 | 52 | 30 | 14.9 | | |
| Specialty (Department) | Emergency | 58 | 32 | 87 | 48.1 | 36 | 19.9 | 15.843 | .045 |
| | ICU | 111 | 26.3 | 178 | 42.2 | 133 | 31.5 | | |
| | Operation | 28 | 27.7 | 34 | 33.7 | 39 | 38.6 | | |
| | Psychiatry | 41 | 24.6 | 67 | 40.1 | 59 | 35.3 | | |
| | Ward | 108 | 27.3 | 169 | 42.7 | 119 | 30 | | |

| Variables | | Keep the records in a more detailed way to protect yourself from malpractice allegations | | | | | | $X^2$ | $P$-Value |
|---|---|---|---|---|---|---|---|---|---|
| | | Always (n = 1,001) | | Sometimes (n = 211) | | Never (n = 55) | | | |
| | | No. | % | No. | No. | % | No. | | |
| Risk of malpractice lawsuit at work | Not at all high | 570 | 77.8 | 122 | 16.6 | 41 | 5.6 | 9.679 | .046 |
| | High | 324 | 80.4 | 65 | 16.1 | 14 | 3.5 | | |
| | Profoundly high | 107 | 81.7 | 24 | 18.3 | 0 | 0 | | |
| Specialty (Department) | Emergency | 152 | 84 | 29 | 16 | 0 | 0 | 50.599 | <.001 |
| | ICU | 330 | 78.2 | 87 | 20.6 | 5 | 1.2 | | |
| | Operation | 89 | 88.1 | 9 | 8.9 | 3 | 3 | | |
| | Psychiatry | 124 | 74.2 | 30 | 18 | 13 | 7.8 | | |
| | Ward | 306 | 77.3 | 56 | 14.1 | 34 | 8.6 | | |

and mental well-being. This finding aligns with other studies showing that many healthcare professionals feel particularly vulnerable to legal action in their daily work (*Maquibar et al., 2023*; *Patynowska et al., 2023*). Perceived risk of malpractice lawsuits can influence how nurses conduct their duties. Studies have shown that higher perceived legal risks can lead to defensive nursing practices, where nurses might over-document or avoid high-risk procedures to protect themselves from potential litigation (*Renkema, Broekhuis & Ahaus, 2014*). This can result in increased workload and stress, potentially affecting the quality of patient care (*Savcı et al., 2020*).

Our results about defensive nursing practices are prevalent among the 1,267 surveyed nurses, reflecting their strategies to mitigate the risk of malpractice allegations. These practices can be broadly categorized into positive and negative defensive behaviors. The survey highlights that positive defensive practices are common. A significant 58.5% of nurses always explain nursing procedures in more detail to patients, and 79% keep more detailed records. These practices are intended to ensure transparency, improve patient understanding, and provide thorough documentation that can be crucial in defending against malpractice claims. Such meticulousness in communication and record-keeping can enhance patient care and build trust, ultimately serving both the nurses' and patients' interests (*National Academies of Sciences, Engineering, and Medicine et al., 2021*). *Bizimana & Bimerew (2021)*, in a study conducted on 121 nurses at two public district hospitals in Burundi, reported that nurses had positive attitudes toward the benefits of high-quality patient record-keeping. Similarly, *Turan & Kaya (2019)*, in their study involving 345 nurses working in inpatient departments in hospitals across Turkey, found that 48.7% of nurses sometimes explained nursing practices in greater detail to protect themselves from malpractice allegations. Regular record-keeping was highlighted as one of the key methods for combating malpractice (*Yılmaz, Polat & Kocamaz, 2014*). Additionally, *Robertson & Thomson (2016)*, in a study conducted across various regions of England, reported that midwives increased the amount of documentation they maintained as a defensive practice.

Despite the emphasis on positive practices, a subset of nurses engage in negative defensive behaviors. The survey indicates that 15.9% of nurses always avoid high-risk procedures, and 13.6% avoid patients they perceive as more likely to file lawsuits. These practices, while aimed at self-protection, can have adverse effects on patient care and access to necessary medical procedures. Avoiding high-risk patients or procedures can lead to disparities in care and negatively impact patient outcomes (*Wade et al., 2022*). At same line, *Turan & Kaya (2019)* stated that 44.1% of them never avoided practices with high complications to guard themselves against malpractice lawsuits. Midwives and nursing personnel practiced both active and passive defensive practices, such as over-investigation, over-treatment, and avoidance of high-risk patients (*Rinaldi et al., 2019*). *Guidera et al. (2012)*, in a study conducted in the United States, reported a 23% decrease in the number of high-risk patients cared for, highlighting the impact of defensive practices on healthcare delivery. Similarly, *Hood, Fenwick & Butt (2010)*, in their study on Australian midwives, found that midwives increased the number of consults with collaborating physicians as a defensive practice, reflecting a strategy to mitigate personal liability in high-risk situations.

The prevalence of both positive and negative defensive practices underscores the need for a balanced approach to nursing practice. Training programs that emphasize effective communication, thorough documentation, and ethical decision-making can help nurses navigate the complexities of patient care without resorting to avoidance strategies. Additionally, institutional support systems that provide legal and emotional support to nurses can reduce the perceived need for negative defensive behaviors.

The role of institutional support systems, particularly first-line managers, is critical in influencing defensive nursing practices. First-line managers serve as key intermediaries between frontline nurses and organizational leadership, shaping workplace environments and setting the tone for workplace culture. Supportive managers play a vital role in mitigating defensive practices by fostering open communication, ensuring fair treatment, addressing workplace violence promptly, and providing emotional and professional support to nursing staff. These actions can reduce nurses' fears of legal risks and workplace pressures, thereby decreasing the likelihood of negative defensive behaviors.

Conversely, a lack of managerial support can exacerbate these fears, increasing nurses' reliance on defensive practices to protect themselves from potential liability or conflict. For example, *Gunawan et al. (2023)* emphasize that managerial support is essential for reducing workplace stress and promoting effective, patient-centered decision-making among nurses. Incorporating leadership training programs for first-line managers that focus on conflict resolution, emotional intelligence, and staff engagement could address the institutional challenges that drive defensive practices. These strategies would not only support nurses but also create a safer, more trusting workplace environment conducive to high-quality care.

The relationship between nurses' socio-demographic characteristics and their practice of explaining nursing procedures in detail to mitigate malpractice allegations reveals several significant trends. Age was significantly associated with the likelihood of explaining procedures in detail ($\chi^2 = 18.124$, $p = .001$). Nurses aged 40–50 years old were the most likely to always provide detailed explanations (74.7%), compared to their younger. This trend suggests that older nurses may have more experience and awareness of the importance of detailed communication in protecting against legal issues (*Hanganu et al., 2022*). Also, Educational qualifications also showed a significant correlation with this practice. Nurses holding bachelor's (60.5%) and postgraduate degrees (51%) were more likely to always explain procedures in detail. This indicates that higher education levels may equip nurses with better communication skills and a deeper understanding of legal implications in nursing practice (*Alandajani et al., 2022*). Furthermore, clinical specialty significantly influenced the practice of detailed explanations ($\chi^2 = 31.065$, $p < .001$). Nurses in emergency (69.1%) and psychiatry departments (50.9%) most frequently reported always explaining procedures in detail. These specialties often involve high-stress environments and complex patient interactions, possibly necessitating more thorough communication to prevent misunderstandings and legal complications (*Al-Hasnawi & Aljebory, 2023*).

According to the relationship between nurses' experiences of violence or legal consequences and their practice of avoiding high-complication procedures to protect

against malpractice allegations. Nurses who had experienced physical workplace violence were significantly more likely to avoid high-complication procedures. This suggests that the trauma and fear resulting from physical violence lead to increased caution and defensive behavior among affected nurses (*Kim, Mayer & Jones, 2021*). Similarly, nurses who had been threatened with legal consequences by patients or their families more frequently avoiding high-complication procedures. The fear of litigation appears to drive nurses towards more conservative practices to minimize the risk of legal action (*Chapman et al., 2008*).

## CONCLUSION

The study identified a significant engagement in positive defensive practices among nurses, such as detailed documentation and thorough explanation of procedures to patients, primarily aimed at legal protection. Despite the prevalence of these positive practices, negative defensive behaviors, such as avoiding high-risk patients and procedures, were also observed, albeit less frequently. The analysis revealed that older nurses and those with higher educational qualifications were more likely to engage in positive defensive practices. Additionally, nurses working in emergency and psychiatry departments reported higher incidences of these practices. Importantly, the study found that nurses who experienced workplace violence or legal threats were more likely to avoid high-complication procedures, indicating a direct link between these adverse experiences and the adoption of defensive practices. These findings highlight the critical need for strategies to support nurses, reduce the reliance on defensive practices, and ultimately ensure better patient outcomes.

### Limitations

Sample representation: The study relied on self-reported data from a convenience sample of nurses, which may not be representative of all nurses in Egypt. Response bias: there is a possibility of response bias, where nurses might have reported practices in a socially desirable manner. Cross-sectional design: the study's cross-sectional nature limits the ability to establish causality between defensive practices and their influencing factors.

### Implications of practices

Policy sevelopment: the findings can inform policy development aimed at reducing legal threats and improving workplace safety for nurses. Training programs: implementing training programs focused on effective communication, legal knowledge, and emotional resilience can help nurses manage their fears and reduce negative defensive practices. Support systems: establishing robust support systems, including legal and psychological support, can help mitigate the factors driving defensive practices and promote a safer and more effective nursing environment.

### Funding

This work was supported by Princess Nourah bint Abdulrahman University Researchers Supporting Project number (PNURSP2025R444), Princess Nourah bint Abdulrahman University, Riyadh, Saudi Arabia. The funders had no role in study design, data collection and analysis, decision to publish, or preparation of the manuscript.

### Grant Disclosures

The following grant information was disclosed by the authors:
Princess Nourah bint Abdulrahman University: PNURSP2025R444.

### Competing Interests

The authors declare there are no competing interests.

### Author Contributions

- Ahmed Zaher conceived and designed the experiments, authored or reviewed drafts of the article, and approved the final draft.
- Yasmine M. Osman performed the experiments, authored or reviewed drafts of the article, and approved the final draft.
- Salwa Sayed conceived and designed the experiments, analyzed the data, authored or reviewed drafts of the article, and approved the final draft.
- Sally Mohammed Farghaly Abdelaliem performed the experiments, prepared figures and/or tables, and approved the final draft.
- Amany Anwar Saeed Alabdullah analyzed the data, prepared figures and/or tables, and approved the final draft.
- Ahmed Hendy analyzed the data, prepared figures and/or tables, and approved the final draft.
- Zainab Attia Abdallah conceived and designed the experiments, authored or reviewed drafts of the article, and approved the final draft.
- Mohammed Musaed Ahmed Al-Jabri performed the experiments, authored or reviewed drafts of the article, and approved the final draft.
- Abdelaziz Hendy conceived and designed the experiments, authored or reviewed drafts of the article, and approved the final draft.

### Ethics

The following information was supplied relating to ethical approvals (i.e., approving body and any reference numbers):

The research was approved by an ethical and research committee affiliated with the faculty of nursing at Modern University for Technology and Information (MTI) with the formal approval number (FAN/128/2023) on December 25, 2023.

### Data Availability

The raw data are available in the Supplementary File.

## Supplemental Information

Supplemental information for this article can be found online at http://dx.doi.org/10.7717/peerj.19005#supplemental-information.

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
