# Peer review of "When safety becomes the priority: defensive nursing practice and its associated factors among nurses in Egypt: a cross-sectional study"

_PeerJ, doi:10.7717/peerj.19005_

## Round 0.1 · original submission · Major Revisions

Please address the comments of both reviewers. In particular, Reviewer 2 has many relevant comments which must be responded to.

·

Basic reporting

no comment

Experimental design

no comment

Validity of the findings

no comment

Additional comments

1. The manuscript is important for the scientific community as the author/s ultimately explored the prevalence and factors associated with defensive nursing practices among nurses in Egypt, considering the impact of workplace violence and legal threats.
The study identified a high engagement in both positive and negative defensive practices among nurses in Egypt. These practices are influenced by factors such as age, education level, and experiences of workplace violence and legal threats. The findings underscored the need for strategies to support nurses, reduce reliance on defensive practices, and ensure better patient outcomes.
2. The title suits the article best. It reveals what the researchers want to convey in the study.
3. The abstract is comprehensive as it includes information that is important for understanding the content of the manuscript.
4. The article about the prevalence and factors associated with defensive nursing practices among nurses in Egypt has appropriate discussions in its different sections, from the abstract; introduction; materials, and methods; to its results and discussions. What transpired during their research conduction was laid down properly by the authors in the different sections of the article which made it clearer and more understandable to its readers.
5. I think the manuscript is scientifically correct as it concludes about high engagement in both positive and negative defensive practices among nurses in Egypt. And they have laid down the different factors that influenced these practices such as age, education level, and experiences of workplace violence and legal threats. Their findings underscored that there is a need for strategies to support nurses, this is to reduce reliance on defensive practices and ensure better patient outcomes.
6. I think the article’s quality in using English is suitable for scholarly communications.
7. The manuscript does not have ethical issues.
8. Arguments are logical and supported by data. The arguments support the conclusion.
9. There are no competing interest issues in this manuscript.
10. I think this is not plagiarized.
11. The references are not all recent and are not formatted using the 7th edition APA formatting which could have made it easier to trace.
12. Minor Revision: (>8-9) – some references should be updated

·

Basic reporting

The key concept of “defensive practice” and “defensive nursing practice” could benefit from further elaboration to provide the necessary context for the reader and to support the rationale for the study. For instance, explaining what constitutes defensive nursing practice in more detail and how it relates to existing literature would strengthen the introduction. I also recommend adhering more closely to the CARS (Create a Research Space) model for the introduction, as developed by John Swales, to ensure a more coherent and structured argument. In its current form, the logic in some parts of the introduction is not entirely clear, particularly regarding the relationship between increasing nurse responsibilities and the issue of restrictive practice. Additional clarification would help avoid any inconsistency in the narrative. Please see the specific comments in the PDF.
There are significant issues with the reference list, which contains numerous typos. For example, one reference cites an author as “Mahmoud Farhat, A.A., S. Ghandour, and H.A.J.P.S.S.J.o.N. Mohamed.” It is highly unlikely the final author has multiple first names, and these initials appear erroneous. The journal names are also missing in all references, which is a critical omission.

Experimental design

The design of the study is solid, particularly the sample size, which appears robust enough to generate meaningful findings. However, I suggest that you further elaborate on this in the discussion section. A clearer discussion of why the sample size is strong would enhance this section.
Additionally, the process of translating the questionnaire should be more rigorously reported. Translation processes, particularly for validated questionnaires, are often complex and require careful documentation. I recommend you to include robust citations, for example among others, Thompson et al. (2024) published an article titled "Forward and Back is Not Enough" (10.3389/frsle.2023.1329405). You should refer to standardized methods for translating questionnaires, as outlined in previously published methodological articles or books. Another example is the article by Shalev et al. (2023) entitled “Bridging Language Barriers in Developing Valid Health Policy Research Tools: Insights from the Translation and Validation Process of the SHEMESH Questionnaire,” which may offer some useful guidance.

Validity of the findings

The findings are of high relevance, but the discussion could benefit from a more in-depth analysis of institutional support systems, particularly the role of first-line managers and their role in defensive nursing practices. This is an important factor that is under-discussed in the current version. Supporting this point with additional references would enhance the depth of your analysis.

Additional comments

In terms of presentation, there are a few points where clarification is needed. For instance, in the introduction, it is unclear what exactly leads patients to adopt a restrictive practice. This inconsistency in logic could confuse readers, so I suggest a more explicit explanation here. Furthermore, some references need further elaboration. For example, when mentioning studies on defensive practice among physicians, a brief description of the study’s context (e.g., “a study from the Netherlands reported...”) would strengthen your argument and give the reader more information to understand the scope of the evidence being cited.

---

## Round 0.2 · accepted · Accept

Dear Dr. Hendy,

Your manuscript has been Accepted for publications. Congratulations!

·

Basic reporting

Meets the criteria suitable for publication.

Experimental design

Meets the criteria suitable for publication.

Validity of the findings

Meets the criteria suitable for publication.

Additional comments

The authors have revised the manuscript per my queries.